# Similarities in Pathogenetic Mechanisms Underlying the Bidirectional Relationship between Endometriosis and Pelvic Inflammatory Disease

**DOI:** 10.3390/diagnostics13050868

**Published:** 2023-02-24

**Authors:** Hiroshi Kobayashi

**Affiliations:** 1Department of Gynecology and Reproductive Medicine, Ms.Clinic MayOne, Kashihara 634-0813, Japan; hirokoba@naramed-u.ac.jp; 2Department of Obstetrics and Gynecology, Nara Medical University, Kashihara 634-8522, Japan

**Keywords:** endometriosis, immune response, microbiota, pelvic inflammatory disease, tubo-ovarian abscess

## Abstract

Background: Endometriosis is a common inflammatory disease characterized by the presence of endometrial cells outside of the uterine cavity. Endometriosis affects 10% of women of reproductive age and significantly reduces their quality of life as a result of chronic pelvic pain and infertility. Biologic mechanisms, including persistent inflammation, immune dysfunction, and epigenetic modifications, have been proposed as the pathogenesis of endometriosis. In addition, endometriosis can potentially be associated with an increased risk of pelvic inflammatory disease (PID). Changes in the vaginal microbiota associated with bacterial vaginosis (BV) result in PID or a severe form of abscess formation, tubo-ovarian abscess (TOA). This review aims to summarize the pathophysiology of endometriosis and PID and to discuss whether endometriosis may predispose to PID and vice versa. Methods: Papers published between 2000 and 2022 in the PubMed and Google Scholar databases were included. Results: Available evidence supports that women with endometriosis are at increased risk of comorbid PID and vice versa, supporting that endometriosis and PID are likely to coexist. There is a bidirectional relationship between endometriosis and PID that shares a similar pathophysiology, which includes the distorted anatomy favorable to bacteria proliferation, hemorrhage from endometriotic lesions, alterations to the reproductive tract microbiome, and impaired immune response modulated by aberrant epigenetic processes. However, whether endometriosis predisposes to PID or vice versa has not been identified. Conclusions: This review summarizes our current understanding of the pathogenesis of endometriosis and PID and discusses the similarities between them.

## 1. Introduction

Endometriosis is an estrogen-dependent inflammatory disease that is characterized by the growth of endometrial glands and stroma located outside of the uterine cavity [1]. This disease can be classified into three variants: ovarian endometrioma, superficial peritoneal disease, and deep infiltrating endometriosis. Of these, ovarian endometrioma is a common disease. Endometriosis affects approximately 10% of reproductive-aged women and is accompanied by pelvic pain and infertility. Endometriosis is associated with a wide array of diseases, such as ovarian cancer, breast cancer, autoimmune diseases, cardiovascular disease, and asthma, compromising the quality of life [2]. Additionally, women with endometriosis have been reported to be at an increased risk of pelvic inflammatory disease (PID) [3]. PID is a common inflammatory disease and is characterized by an infection of the upper genital tract (e.g., the uterus, fallopian tubes, ovaries, and pelvic peritoneum) [4]. PID is often believed to be a sexually transmitted disease, but bacterial vaginosis (BV) is also associated with an increased risk of PID [4]. Ovarian abscesses (OA) and tubo-ovarian abscesses (TOA) are serious complications of PID and prove difficult to treat due to resistance to treatments and several complications [5]. They often affect the female reproductive tract and other adjacent pelvic organs or structures, leading to adverse reproductive outcomes [5]. Short- and long-term complications and sequelae include infertility, ectopic pregnancy, chronic pelvic pain, and recurrent infections due to dense pelvic adhesions and macroscopic anatomic and morphological abnormalities [6]. The available clinical and epidemiological evidence supports that PID and TOA occur more frequently in women with endometriosis than in those without endometriosis [7,8]; however, it is unclear if women with endometriosis are more prone to PID or vice versa.

This review aims to investigate the mechanisms underpinning the bidirectional relationship between endometriosis and PID and to discuss their similarities. Our current understanding of anatomical, morphological, microbiological, immunological, and epigenetic alterations would provide new insights into a potential association between the two diseases.

## 2. Materials and Methods

### 2.1. Search Strategy and Selection Criteria

A computerized literature search was performed to identify relevant studies in English. The PubMed and Google Scholar electronic databases were searched for studies published between January 2000 and October 2022. The search terms included endometriosis, immune response, microbiota, pelvic inflammatory disease, and tubo-ovarian abscess. In the search strategy, these keywords were combined with the Boolean operators AND and OR, as described in Table 1. Inclusion criteria included the publication of original studies and reference lists in review articles. The references of each article were searched to identify potentially relevant studies. Figure 1 shows the first identification phase which includes records identified through a database search. Terms in the titles and abstracts were searched during the first screening. During the second screening, duplicates were removed, and titles and abstracts were read to remove inappropriate papers. The final eligibility phase included full-text articles for analysis after excluding those wherein detailed data could not be extracted.

### 2.2. Selection of Studies

The literature search on PubMed and Google Scholar provided 554 records (Figure 1). After removing unsuitable (titles and abstracts irrelevant to the topic) and duplicate articles, we obtained 103 records, of which 82 were excluded, and 21 met the selection criteria.

This figure shows the number of articles identified by keyword combinations and the number of records identified through database searching, records after duplicate removal, records screened, removal of inappropriate articles by reading full-text articles, and full-text articles assessed for eligibility.

## 3. Results

### 3.1. A Bidirectional Relationship between Endometriosis and PID

#### 3.1.1. PID

PID is defined as an infection of the upper genital tract, including endometritis, salpingitis, peritonitis, OA, and TOA [9,10]. Approximately one-third of patients with PID manifest TOA [10,11]. The common bacterial organisms isolated and identified were sexually transmitted organisms (e.g., Neisseria gonorrhoeae, Chlamydia trachomatis, Mycoplasma genitalium, and Trichomonas vaginalis); anaerobic and aerobic organisms associated with BV (e.g., Atopobium vaginae, Mycoplasma genitalium, Gardnerella vaginalis, Prevotella species, Sneathia, and Megasphaera); and organisms isolated from the enteric or respiratory tracts (e.g., Bacteroides, Escherichia coli, aerobic Streptococcus, or Haemophilus influenza) [8,9,12,13]. In some cases, it can be difficult to identify the causative pathogens of PID. The pathogenic bacteria frequently isolated from clinical specimens in women with TOA, a severe form of PID, are facultative anaerobic bacteria (e.g., Escherichia coli and Enterococcus spp) and obligate anaerobic bacteria (e.g., Bacteroides fragilis) [5,10,14,15,16]. Uncommon pathogens from other body sites can also reach the ovaries through the bloodstream [8]. In addition, TOA in women with ovarian endometrioma are frequently polymicrobial, complicated by sexually transmitted organisms (e.g., Neisseria gonorrhoeae and Chlamydia trachomatis) and obligate anaerobic bacteria (e.g., Bifidobacterium, Bacteroides, Eubacterium, and Clostridium) [17,18]. TOA infections can also be caused by a multitude of different microbes from BV (e.g., Mycoplasma genitalium, Gardnerella vaginalis) and normal intestinal commensal organisms (e.g., Ruminococcus gnavus, a Gram-positive anaerobe) [17,18]. The development of TOA in women with ovarian endometrioma has been reported to be associated with lower genital tract infections and endometriotic cyst rupture [19]. Therefore, a mixed infection caused by numerous BV-associated bacterial organisms that can infect the upper genital tract may predispose women with ovarian endometrioma to abscess formation [17]. Much of the literature discusses ovarian endometrioma and PID, but very little focus has been placed on peritoneal endometriosis. Therefore, it is unknown whether peritoneal endometriosis is associated with PID.

A wide array of pathogenetic mechanisms and risk factors have been proposed for the development and progression of PID or TOA: young age (<25 years), the size of the abscess >6 cm, multiple sexual partners, frequent partner turnover, a previous history of PID, a previous history of intrauterine device (IUD) insertion, a previous laparotomy, assisted reproductive technology, oocyte retrieval, structural genital anomalies, drug consumption, cigarette smoking, diabetes, and some immune deficiency diseases [10,20,21,22,23,24,25,26]. However, it is rarely (4%) reported to occur in virgins, even at a young age [27]. A comprehensive review of the literature identified a study including 16 virgins in 2019 [27]. TOA in virgins is often secondary to associated comorbidities, such as congenital malformations of the female genitourinary tract, obstructed hemivagina with a renal anomaly, bowel malrotation, complicated appendicitis, bacteremia caused by wound sepsis or dental procedures, and prior abdominopelvic surgery [8,16,27,28,29]. Thirteen (81%) of 16 virgins had these relevant comorbidities [27]. Therefore, these comorbidities, including genitourinary tract anomalies, can predispose virgins to TOA.

#### 3.1.2. Are Women Having PID Prone to Endometriosis?

This subsection focuses on the prevalence or incidence of endometriosis in women with PID to investigate whether PID may predispose women to endometriosis. In a recent study, 81 (69.2%) of 117 patients with PID had TOA, 59 (72.8%) of whom were complicated with ovarian endometrioma, indicating that about half (50.4%) of PID patients experienced concurrent endometriosis [30]. Furthermore, Elizur et al. reported that concurrent endometriosis was diagnosed in 21 (14.2%) of 148 patients with PID or TOA [31]. In addition, women with PID had a 3- or 4-fold increased risk of developing endometriosis compared to women without PID [32] or the general population [18,33], respectively. Therefore, endometriosis is common in patients with PID or TOA, with incidences ranging from 14.2% to 72.8%. Available evidence supports that patients with PID are at increased risk of comorbid endometriosis.

#### 3.1.3. Are Women Having Endometriosis Prone to PID?

Next, the author summarizes whether patients with endometriosis are at increased risk of comorbid PID. In a retrospective cohort study, Grammatikakis et al. identified 21 (2.9%) patients with PID among 720 women who underwent surgery for ovarian endometrioma, suggesting that the prevalence of PID in women with endometriosis is significantly higher than in the general population [7]. These patients had an average age of 31 years (range: 21–39 years), no laterality of the ovary, and a mean diameter of the endometriotic cyst of 3.5 cm [7]. Furthermore, a recent retrospective study also found that 196 (63%) of 311 patients who underwent surgery for PID had a history of endometriosis, demonstrating that patients with PID experience a higher prevalence of endometriosis compared to the general population or the population in tertiary care [34]. Similar results have been obtained in several studies. Women with endometriosis are more likely to develop TOA than women without endometriosis, with risk factors being younger age [35] and more advanced stage [36,37]. The potential risks for developing TOA in women with ovarian endometrioma were reproductive tract infections and the spontaneous rupture of endometriotic cysts [19]. Furthermore, the rate of recurrent PID in endometriosis patients with TOA was as high as 21.4%, indicating that endometriosis is an independent risk factor for PID recurrence [33]. Coexisting endometriosis was the potentially important risk factor that influences TOA postoperative recurrence [20,33,38]. In addition, pelvic abscess formation following oocyte retrieval during an in vitro fertilization (IVF) cycle is a rare complication (0.1–0.4%) but is known to be at increased risk in patients with endometriosis [7,31,39,40,41,42,43]. In contrast, one retrospective study showed that oocyte retrieval was not linked to the development of TOA in women with endometriosis [37]. Despite some inconsistent evidence in the literature, much data support the association between IVF treatment and TOA risk. The treatment strategies for PID patients with endometriosis are essentially the same as those for women without endometriosis [8]. However, the treatment of PID in patients with endometriosis is often complicated by refractory to antibiotic treatment, increased need for surgery, multiple intraoperative and postoperative complications, and prolonged hospitalization [7,8,31,35,37]. Overall, an increasing body of evidence suggests that endometriosis may be associated with an increased risk of incidence, severity, and recurrence of PID, as well as difficulty in treatment selection. Taken together, TOA can be caused by a complex combination of comorbid endometriosis, surgical procedures associated with IVF treatment, and treatment resistance.

### 3.2. A Pathophysiological Mechanism Coupling Endometriosis and PID

In the previous section, I showed that endometriosis and PID are likely to coexist. This section summarizes why women with endometriosis are at increased risk for PID, with a particular focus on the distortions in the morphology of the pelvic organs (i.e., anatomical distortions), the periodic bleeding of ectopic lesions, alterations of the reproductive tract microbiota, and impaired immune systems.

#### 3.2.1. Distortion of the Pelvic Anatomy Favoring Bacteria Proliferation

Several potential causes support the increased risk of PID in women with endometriosis, one of which may be the distorted anatomy favoring bacteria proliferation. Endometriosis is characterized by abnormal activation of inflammation, macrophage infiltration, and angiogenesis that leads to surrounding tissue adhesion and fibrosis formation, thereby distorting the pelvic organs [8,44]. PID typically causes anatomical distortions of the ovaries, fallopian tubes, and retroperitoneum through inflammation caused by bacterial infection, resulting in a frozen pelvis. Despite clinical differences, endometriosis and PID share some common characteristics, such as inflammation, adhesions, and distortion of the pelvic organs. It is well known that a history of various intrauterine manipulations and pelvic surgery or retrograde menstruation caused by congenital obstructive abnormalities predisposes to PID or endometriosis, respectively [30,45]. The distorted anatomy of the pelvic organs provides an optimal environment for bacteria and may favor the growth of diverse bacteria. Therefore, similarities in morphological changes in an anatomical structure may be associated with an increased risk of PID in women with endometriosis.

#### 3.2.2. Nutrients from Old Blood in Endometriotic Lesions

The second cause that explains the risk may be old blood within the endometriotic cyst or in the peritoneal cavity. Lactic acid produced by Lactobacillus acidifies the vagina, thereby providing a prompt defense against infections of pathogenic bacteria [46]. The menstrual blood increases the pH of the vagina and uterine cavity negatively, which affects commensal microbiota and is conducive to pathogenic bacterial proliferation, contributing to an increase in the risk of bacterial infection [46]. An increase in vaginal fluid pH is known to be associated with BV [46]. Furthermore, degenerated old blood within endometriotic cysts and retrograde menstrual blood that enters the peritoneal cavity are suitable culture mediums for bacteria and can promote the growth of pathogenic microorganisms. Blood supplies the bacteria with oxygen, nutrients, and energy substrates such as glucose, lactate, and proteins. In fact, blood agar plates are used in the clinical laboratory for the identification of bacteria [47]. Various bacteria such as Escherichia coli, Staphylococcus aureus, Streptococcus pyogenes, and Streptococcus pneumoniae form colonies on blood agar plates [47]. Therefore, degenerated old blood in women with endometriosis favors the growth of pathogenic bacteria. Periodic bleeding in ectopic lesions creates a nutrient-rich environment that favors the growth of diverse bacteria and may facilitate the spread of infection [36]. Taken together, these findings suggest that women with endometriosis are more prone to infections and more likely to develop PID than normal individuals or women without endometriosis.

In contrast, menstrual blood contributes to the defense against an array of microorganisms transported by sperm [48]. Menstrual blood discharge is rich in hemocyanins, bactericidal peptides generated from hemoglobin [49]. Hemocidins exhibit potent antimicrobial activity toward some strains of bacteria, especially Gram-negative bacteria [49]. However, normal menstrual discharge and degenerated old blood in endometriotic cysts may differ in their physiological properties. It is not known whether hemocyanins in endometriotic cyst fluid are sufficient to stop bacterial growth.

#### 3.2.3. Alterations of the Gut and Reproductive Tract Microbiome

Thirdly, alterations in the reproductive tract microbiomes in patients with endometriosis may be intricately linked with the development of PID or TOA [50]. Microbiota is the diverse collection of vital microorganisms that live in symbiosis with humans [51,52]. They are found in the skin, mouth, gastric, gut, and female reproductive tract. The commensal microbiota regulates inflammatory, immune, and metabolic functions, which constitute a protective barrier against pathogen infections and eliminate foreign intruders or pathogenic bacteria [51]. In fact, the hygiene hypothesis states that commensal bacteria promote early protection against inflammatory diseases and allergies [52]. On the other hand, alteration of the composition and function of the microbiota, i.e., dysbiosis, is linked with various conditions, such as inflammatory bowel disease, allergies, autoimmunity, reproductive disorders, and cancer through the stimulation of innate immune responses and the induction of several inflammatory cascades [53].

The microbiota of women with endometriosis is characterized by reduced Lactobacillus abundance in the lower genital tract and diverse flora of facultative and anaerobic organism populations, such as Gardnerella, Streptococcus, Enterococci, Escherichia coli, and Mollicutes in the upper genital tract, peritoneum, and endometriotic lesions [3,54]. Pathogenic organisms associated with BV ascend from the bottom to the top of the reproductive tract and may predispose to PID and severe forms of abscess formation. Therefore, exposure to infection caused by pathogenic microorganisms in the lower and upper reproductive tract and the peritoneal cavity is a hallmark of endometriosis [3,55]. In addition to abnormalities in the reproductive tract microbiota, dysbiosis in the gut microbiota has been identified in animal models, such as mice and nonhuman primates (e.g., rhesus monkeys), and women with endometriosis [56,57]. There is evidence from animal models showing that the transplantation of endometrial fragments increases specific gut microbes, such as obligatory anaerobic bacteria (e.g., Ruminococcaceae, Bifidobacterium, and Parasutterella generae) [56]. Therefore, endometriotic lesions cause changes in the gut and reproductive tract microbial abundance, composition, and diversity, promoting the further growth of pathogenic bacteria [54]. Indeed, dysbiosis of the reproductive tract microbiota in patients with BV or PID has been reported to lead to multiple detrimental reproductive outcomes, including infertility [58].

Today it is well known that patients with endometriosis are constantly exposed to inflammation. Increased numbers of Escherichia coli and higher levels of bacterial endotoxin are observed in the menstrual blood of women with endometriosis compared with control women, leading to the Toll-like receptor 4 (TLR-4)-mediated proliferation of endometriotic lesions [3,59,60]. Lipopolysaccharide (LPS), a cell-wall component (endotoxin) produced by Gram-negative bacteria, activates nuclear factor kappa B (NF-κΒ) and its downstream target, cyclooxygenase-2 (Cox-2) [61]. Prostaglandin E2 (PGE2), a downstream product of Cox-2, generates the inflammatory microenvironment. The PGE2 level in the menstrual blood of women with endometriosis has been reported to be 2–3 times higher than in serum or peritoneal fluid [59]. PGE2 mediates bacterial growth in women with endometriosis via inflammatory responses and proinflammatory cytokine production [59]. PGE2 is also known to inhibit the process of bacterial killing by macrophages to attenuate antibacterial immunity [62]. In addition, Noh et al. reported that Ureaplasma urealyticum infection is implicated in the development of pelvic endometriosis via the TLR-2 signaling pathway in the in vivo mouse model [63]. This animal study showed that bacterial infection can accelerate the progression of endometriosis. Taken together, the reproductive tract microbial abundance, composition, and diversity are hallmarks of endometriosis, resulting in a decrease in beneficial microbes and an increase in harmful microbes [54]. This microbiota dysbiosis induces pelvic inflammation and bacterial growth possibly through the TLR-mediated activation of NF-κΒ and macrophage dysfunction, leading to the development and progression of PID and, ultimately, TOA. Although it is currently unknown whether endometriosis induces changes in microbial diversity and composition or vice versa, endometriosis and PID are closely related pathophysiologically.

#### 3.2.4. Impaired Immune Systems

Fourthly, the strong association between endometriosis and PID suggests that they may share a common biological basis in pathogenesis. This subsection focuses on the aberrant activation of the immune system between endometriosis and PID and discusses whether endometriosis predisposes to PID. Natural killer cells, macrophages, mast cells, neutrophils, dendritic cells, and Tregs have been identified in the endometrium as immune cells associated with physiological processes, such as the menstrual cycle [64]. The vital role of these immune cells includes endometrial protection, decidualization, embryo implantation, placentation, the process of repeated tissue breakdown, repair, and regeneration during the menstrual cycle, and scavenging menstrual debris [64]. There is evidence that different types of immune cells have aberrant functions in women with endometriosis; alterations to and the complex interplay of innate and adaptive immune cells, i.e., the dysregulation of the immune system, have been implicated in the pathogenesis of endometriosis [65,66]. A broad range of alterations in immune cell numbers, distributions, and functions (e.g., increased number and activation of peritoneal and endometrial macrophage phenotypes, altered dynamics in T-cell reactivity and NK cytotoxicity, dysregulation of effector functions in the T lymphocytes, an imbalance in T helper cell subsets (Th1/Th2/Th17), and an increased proportion of Treg cells) and changes in localized and systemic inflammatory mediator profiles have been reported in the serum, peritoneal fluids, and eutopic and ectopic endometrium of women with endometriosis [64,67,68,69,70,71]. Treg cells in the peritoneal cavity are elevated in women with advanced endometriosis, allowing ectopic endometriotic cell implantation, survival, proliferation, and progression [66]. Treg cells also contribute to immune evasion, causing further persistent inflammation via the decreased clearance of ectopic endometrial cells [66,72,73]. On the other hand, women with sexually transmitted organisms-induced inflammation, a typical case of PID, are characterized by a disproportionate influx of innate immune cells and impaired innate immunity (e.g., the activation of macrophages and their phenotypes) and alterations of T-cell activation pathways (e.g., T-cell-mediated production of interferon-γ, altered dynamics in T-cell reactivity and NK cytotoxicity) [74]. The development of such PID is associated with immune dysfunction by innate lymphoid cells. The aberrant activation of innate lymphoid cells is also involved in the progression of endometriotic lesions [75]. Therefore, an already existing endometriosis with altered immune cell profiles, imbalances in immune cell function, and compromised immunosurveillance could be involved in the development and pathogenesis of PID [8,35,54]. We believe that a pathophysiological similarity between these two diseases is the defective immune system, such as disruption of the beneficial symbiosis between host and commensal microbes and incomplete defense against invading pathogenic microbes.

#### 3.2.5. The Immune Landscape Modulated by Epigenetic Factors

Finally, this subsection briefly summarizes whether the epigenetic modifications may predispose women with endometriosis toward PID or vice versa. The immune function across endometriosis is largely modulated by selective epigenetic reprogramming. There are fundamental changes in gene expression through aberrant DNA methylation, histone modifications, and the altered expression of non-coding microRNAs in the eutopic endometrium in patients with endometriosis compared to a normal healthy endometrium [71]. Endometriosis is characterized by complex epigenetic modifications that alter the T-cell landscape [66]. Epigenetic modifications play critical roles in the development, activation, and differentiation of CD4+ and CD8+ T cells, the maintenance of Treg cells to ensure the stable expression of Foxp3, and the memory formation of NK cells [71]. Treg cells may modulate the local host-defense mechanism against endometriotic cell proliferation [76] and bacterial invasion [77]. Consequently, epigenetic modifications induce alterations in the immune cell function, number, populations, and phenotypes, promoting the development of endometriosis and bacterial infection. An impaired immune defense in patients with concurrent endometriosis can lead to the exacerbation of PID, even influencing the development of TOA.

Furthermore, it is well known that the hormonal landscape in endometriosis is modulated by epigenetic modifications, leading to estrogen dependence and progesterone resistance [71]. Given that progesterone suppresses endometrial inflammation via the downregulation of inflammatory cytokine and chemokine transcripts, epigenetic modifications to progesterone receptors and their targets result in a proinflammatory phenotype [78]. In addition, an increased ratio of ERβ-to-ERα due to ERβ hypomethylation represents a molecular signature of inflammation [79]. Endometriosis-specific epigenetic alterations create a beneficial environment for successful bacterial growth and spread through both aberrant immune function and altered sex steroid hormone responsiveness [71,80]. Persistent inflammation due to the elevation of proinflammatory cytokines and disturbed cytokine profiles is thought to be an important trigger of PID exacerbations [71,78,79,80]. Epigenetic modifications help to explain why women with endometriosis are more prone to PID. Conversely, an infection can also trigger epigenetic changes that influence the development of endometriosis [3]. For example, sepsis-associated epigenetic modifications have been reported to be associated with enhanced host–pathogen interaction, impaired macrophage function, prolonged immunosuppression, exaggerated inflammation, impaired mitochondrial energy metabolism, and deficient wound healing [81]. An epigenetic link between endometriosis and PID has been suggested, but research is still in its infancy.

## 4. Discussion

This review summarizes the pathophysiological mechanisms that explain the bidirectional relationship between endometriosis and PID and discusses their similarities. Endometriosis and PID are common inflammatory disorders. The accurate and early detection of PID is crucial for the selection of optimal care and better prognosis because endometriosis with TOA is often a difficult-to-treat disorder [10,11]. Some studies revealed that endometriosis is associated with an increased risk of incidence, severity, complications, treatment failure, and recurrence of PID or TOA [7,36,37]. Conversely, patients with PID are more likely to develop endometriosis, with a survey of the National Health Insurance Research Database showing that individuals with PID had a 3-fold increased risk of developing endometriosis compared with an age-matched control group including patients without PID [32]. Although studies are still limited in number, endometriosis predisposes women to PID and vice versa, with these two conditions influencing each other. A causal relationship between endometriosis and PID remains unclear, but both diseases are likely to coexist. Therefore, this article reviews our current understanding of the shared pathogenesis of endometriosis and PID and highlights the following similarities: (1) the distorted anatomy favorable to bacteria proliferation, (2) periodic hemorrhage of the endometriotic focus, (3) alterations to the gut and reproductive tract microbiome, and (4) impairment of the immune response modulated by epigenetic factors.

The author first focuses on the anatomical similarities between endometriosis and PID. In more advanced stages, both diseases commonly cause severe adhesions and fibrosis [8,30,44,45]. Adhesions secondary to inflammation associated with endometriosis readily facilitate the spread of pathogens to the pelvic cavity. The similarities in the dramatic anatomical changes within the pelvic cavity are thought to be the reason why endometriosis often coexists with PID or TOA [30,45]. The second pathophysiological mechanism may be the bloody content of the endometrioma or in the peritoneal cavity, which favors bacterial growth. Pooled blood contains nutritional components that promote bacterial growth, suggesting that endometriosis is prone to PID, particularly TOA [36]. Surgical procedures (e.g., IVF treatments) in such settings also contribute to infection and abscess formation. Thirdly, the altered microbial abundance, composition, diversity, and function in the reproductive tract of women with endometriosis may be associated with the development and pathogenesis of PID [3,54,55]. Such dysbiosis has negative impacts on immune function including disruptions to the immune pathway, aberrant expressions of immune mediators, attenuated immunosurveillance, and disruption to the immune defense system, all of which may facilitate the development of endometriosis [54]. Over time, this immune dysregulation can progress into a chronic state of inflammation and reduced pathogen resistance, which may drive the vicious cycle of PID onset and progression. Finally, such immune dysfunction can be mediated by endometriosis-specific epigenetic modifications, altering the Treg cell abundance, affecting defense mechanisms to combat pathogens, and increasing the risk of PID, and ultimately, TOA [66,72,73]. Figure 2 illustrates the pathophysiological mechanisms underlying the bidirectional relationship between endometriosis and PID. There are at least two possibilities of a potential model: (1) ascending genital tract infections with exogenous pathogenic bacteria may cause TOA via the secondary infection of endometriotic cysts (Figure 2A) and (2) microbiota dysbiosis associated with endometriosis may trigger the development of TOA (Figure 2B). The former implies that endometriosis is incidentally found in PID, while the latter suggests that endometriosis may predispose to PID. However, at present, there is no direct evidence to prove a bidirectional relationship between them. In the latest article published in January 2023, Kitaya and Yasuo [82] suggested substantial commonality between endometriosis and chronic endometritis in terms of immunological, inflammatory, and infectious aspects and discussed a novel antibiotic strategy for the prevention and treatment of endometriosis. Recent advances in omics technologies may greatly contribute to the elucidation of the mechanistic underpinnings of the spectrum of endometriosis, including endometritis and pelvic inflammation. Taken together, clinicians should always be aware that endometriosis may have pelvic inflammation as an underlying molecular mechanism and vice versa to provide the best care for both diseases.

In conclusion, this review summarizes that endometriosis and PID are likely to coexist and that PID, especially TOA, is a progressive disease caused by a complex combination of anatomical and environmental factors (e.g., an increase in opportunistic pathogens), immunological, epigenetic factors (e.g., altered dynamics in T-cell reactivity), surgical procedures (e.g., IVF treatment), and the presence of coexisting endometriosis. In particular, the dysbiosis of reproductive tract microbiota may predispose women with endometriosis to PID through a broad range of immune dysfunction, including the reduced capability of eliminating pathogenic bacteria.

## Figures and Tables

**Figure 1 diagnostics-13-00868-f001:**
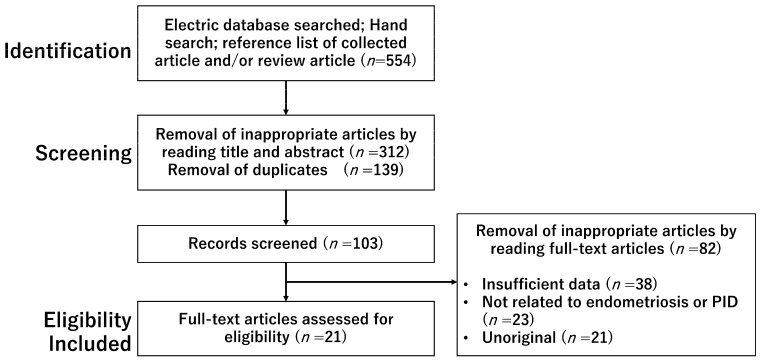
The number of articles identified by searching for keyword combinations.

**Figure 2 diagnostics-13-00868-f002:**
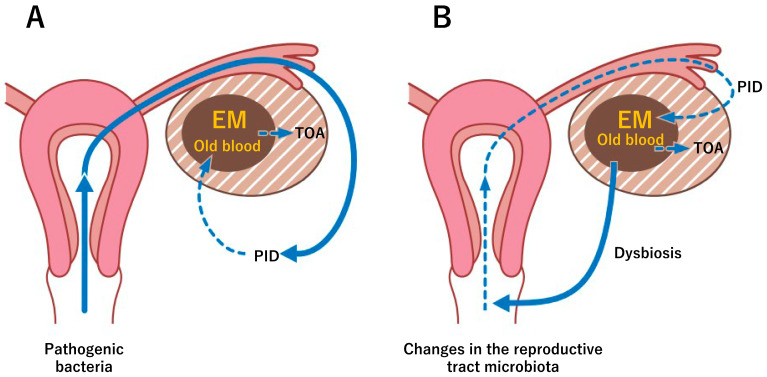
Pathophysiological mechanisms underlying the bidirectional relationship between endometriosis and PID. (**A**) Ascending genital tract infection causes PID, followed by secondary infection of endometriotic cysts, and ultimately TOA. (**B**) Endometriosis-associated microbiota dysbiosis induces pathogenic bacterial growth and pelvic inflammation, leading to secondary infections in endometriotic cysts and then the development of TOA. Bold arrows indicate the first trigger of infection, dotted arrows indicate the second infection.

**Table 1 diagnostics-13-00868-t001:** The search strategy.

Search Mode	The Keyword and Search Term Combinations
Search term 1	endometriosis OR endometrioma OR ovarian endometrioma
Search term 2	pelvic inflammatory disease
Search term 3	immune response OR immune system OR innate OR adaptive
Search term 4	microbiota OR microbiome OR commensal OR pathogenic
Search term 5	tubo-ovarian abscess OR ovarian abscess
Search	Search term 1 AND Search term 2
	Search term 1 AND Search term 2 AND Search term 3
	Search term 1 AND Search term 2 AND Search term 4
	Search term 1 AND Search term 2 AND Search term 5
	Search term 1 AND Search term 3 AND Search term 4
	Search term 2 AND Search term 3 AND Search term 5
	Search term 1 AND Search term 5
	Search term 2 AND Search term 5
	Search term 1 AND Search term 2 AND Search term 5

## Data Availability

No new data were created.

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
