# Peer review of "Similarities in Pathogenetic Mechanisms Underlying the Bidirectional Relationship between Endometriosis and Pelvic Inflammatory Disease"

_diagnostics, 2023, doi:10.3390/diagnostics13050868_

Round 1

Reviewer 1 Report

1.     Provide rationale for rejection of manuscripts from further analyses. Next to obvious “duplicates” the author removed “unsuitable” manuscripts. Please define them.

2.     Please define types of endometriosis covered in the review. Does ovarian or peritoneal endometriosis has the same associations with PID?

3.     The chapter 3.2.2. “Nutriens from…” lacks mechanistic discussion. Further molecular parameters (gene expression, microbiota richness) are missing to discuss this aspect

4.     The reviewer is missing implications to diagnostics (or differential diagnostics recommendations) in the discussion.

Author Response

Answer to the reviewers

Manuscript ID: diagnostics-2233934
Type of manuscript: Review
Title: Title: Similarities in pathogenetic mechanisms underlying bidirectional relationship between endometriosis and pelvic inflammatory disease
Authors: Hiroshi Kobayashi *

Dear Editor in Chief:

Thank you and the reviewers for the thoughtful comments and helpful suggestions on my manuscript “Similarities in pathogenetic mechanisms underlying bidirectional relationship between endometriosis and pelvic inflammatory disease” (manuscript ID: diagnostics-2233934), authored by Hiroshi Kobayashi. I have carefully considered each of the comments, made every effort to address the concerns raised, and applied corresponding revisions to the manuscript. 
The detailed, point-by-point responses to the reviewer comments are given below, whereas the corresponding revisions are highlighted to my manuscript within the document. 
I believe that my manuscript has been considerably improved as a result of these revisions, and hope that the revised manuscript is acceptable for publication in Diagnostics.
I would like to thank you once again for your consideration of my work and inviting me to submit the revised manuscript. I look forward to hearing from you.

With best regards,
Hiroshi Kobayashi, M.D., Ph.D.
Department of Gynecology and Reproductive Medicine, Ms.Clinic MayOne, Kashihara, Nara 634-0813, Japan
Department of Obstetrics and Gynecology, Nara Medical University, Kashihara, Nara 634-8522, Japan.
Tel: +81 744 29 8877
Fax: +81 744 23 6557
E-mail: hirokoba@naramed-u.ac.jp

Point-by-point responses to reviewer comments

Reviewer 1
1.     Provide rationale for rejection of manuscripts from further analyses. Next to obvious “duplicates” the author removed “unsuitable” manuscripts. Please define them.
Response 1:
In section 2.2, changed the following sentence:
The literature search on PubMed and Google Scholar provided 554 records (Figure 1); after removing unsuitable (titles and abstracts irrelevant to the topic) and duplicate articles, we obtained 103 records, of which 82 were excluded, and 21 met the selection criteria.

2.     Please define types of endometriosis covered in the review. Does ovarian or peritoneal endometriosis has the same associations with PID?
Response 2:
I added the following sentence to the end of the first paragraph in Section 3.1.1:
Much of the literature discusses ovarian endometrioma and PID, but very little focus has been made on peritoneal endometriosis. Therefore, it is unknown whether peritoneal endometriosis is associated with PID.

3.     The chapter 3.2.2. “Nutriens from…” lacks mechanistic discussion. Further molecular parameters (gene expression, microbiota richness) are missing to discuss this aspect
Response 3:
Subsection 3.2.2 summarizes the presence of old blood as a nutrient for bacteria. A discussion on mechanisms (expression of molecular parameters and microbiota abundance) is presented in subsection 3.2.3. I would appreciate it if you would check subsection 3.2.3.

4.     The reviewer is missing implications to diagnostics (or differential diagnostics recommendations) in the discussion.
Response 4:
I added the following sentence to the end of the second paragraph in the Discussion section:
Taken together, clinicians should always be aware that endometriosis may have pelvic inflammation as an underlying molecular mechanism and vice versa to provide the best care for both diseases.

Reviewer 2 Report

Very good manuscript. 

I have only one suggestion.

Endometriosis is thought to be also related to chronic endometritis, an inflammatory disorder of the uterine lining that is associated female infertility.

I recommend a discussion on this matter and quotation of the following recent paper: Kitaya, K.; Yasuo, T. Commonalities and Disparities between Endometriosis and Chronic Endometritis: Therapeutic Potential of Novel Antibiotic Treatment Strategy against Ectopic Endometrium. International Journal of Molecular Sciences 2023, 24, 2059.

Author Response

Answer to the reviewers

Manuscript ID: diagnostics-2233934
Type of manuscript: Review
Title: Title: Similarities in pathogenetic mechanisms underlying bidirectional relationship between endometriosis and pelvic inflammatory disease
Authors: Hiroshi Kobayashi *

Dear Editor in Chief:

Thank you and the reviewers for the thoughtful comments and helpful suggestions on my manuscript “Similarities in pathogenetic mechanisms underlying bidirectional relationship between endometriosis and pelvic inflammatory disease” (manuscript ID: diagnostics-2233934), authored by Hiroshi Kobayashi. I have carefully considered each of the comments, made every effort to address the concerns raised, and applied corresponding revisions to the manuscript. 
The detailed, point-by-point responses to the reviewer comments are given below, whereas the corresponding revisions are highlighted to my manuscript within the document. 
I believe that my manuscript has been considerably improved as a result of these revisions, and hope that the revised manuscript is acceptable for publication in Diagnostics.
I would like to thank you once again for your consideration of my work and inviting me to submit the revised manuscript. I look forward to hearing from you.

With best regards,
Hiroshi Kobayashi, M.D., Ph.D.
Department of Gynecology and Reproductive Medicine, Ms.Clinic MayOne, Kashihara, Nara 634-0813, Japan
Department of Obstetrics and Gynecology, Nara Medical University, Kashihara, Nara 634-8522, Japan.
Tel: +81 744 29 8877
Fax: +81 744 23 6557
E-mail: hirokoba@naramed-u.ac.jp

Point-by-point responses to reviewer comments

Reviewer 2
Very good manuscript. 
I have only one suggestion.
Endometriosis is thought to be also related to chronic endometritis, an inflammatory disorder of the uterine lining that is associated female infertility.
I recommend a discussion on this matter and quotation of the following recent paper: Kitaya, K.; Yasuo, T. Commonalities and Disparities between Endometriosis and Chronic Endometritis: Therapeutic Potential of Novel Antibiotic Treatment Strategy against Ectopic Endometrium. International Journal of Molecular Sciences 2023, 24, 2059.
Response 1:
I added the following sentence to the end of the second paragraph in the Discussion section:
In the latest article published in January 2023, Kitaya and Yasuo suggested substantial commonality between endometriosis and chronic endometritis in terms of immunological, inflammatory, and infectious aspects and discussed a novel antibiotic strategy for prevention and treatment of endometriosis. Recent advances in omics technologies may greatly contribute to the elucidation of mechanistic underpinnings of the spectrum of endometriosis, including endometritis and pelvic inflammation.